# Evolution of Rabies Virus Isolates: Virulence Signatures and Effects of Modulation by Neutralizing Antibodies

**DOI:** 10.3390/pathogens11121556

**Published:** 2022-12-19

**Authors:** Juliana Amorim Conselheiro, Gisely Toledo Barone, Sueli Akemi Taniwaki Miyagi, Sheila Oliveira de Souza Silva, Washington Carlos Agostinho, Joana Aguiar, Paulo Eduardo Brandão

**Affiliations:** 1Laboratory of Diagnostics of Zoonosis and Vector-borne Diseases (LabZoo), Zoonosis Surveillance Division, Health Surveillance Coordination, Municipal Health Department, São Paulo 02031-020, SP, Brazil; 2Department of Preventive Veterinary Medicine and Animal Health, School of Veterinary Medicine, University of São Paulo, São Paulo 05508-270, SP, Brazil

**Keywords:** rabies, evolution, quasispecies, signature, antibodies

## Abstract

*Lyssavirus rabies* (RABV) is an RNA virus and, therefore, is subject to mutations due to low RNA polymerase replication fidelity, forming a population structure known as a viral quasispecies, which is the core of RNA viruses’ adaptive strategy. Under new microenvironmental conditions, the fittest populations are selected, and the study of this process on the molecular level can help determine molecular signatures related to virulence. Our aim was to survey gene signatures on nucleoprotein and glycoprotein genes that might be involved in virulence modulation during the in vitro evolution of RABV lineages after serial passages in a neuronal cell system with or without the presence of neutralizing antibodies based on replicative fitness, in vivo neurotropism and protein structure and dynamics. The experiments revealed that amino acids at positions 186 and 188 of the glycoprotein are virulence factors of *Lyssavirus rabies*, and site 186 specifically might allow the attachment to heparan as a secondary cell receptor, while polymorphism at position 333 might allow the selection of escape mutants under suboptimal neutralizing antibodies titers.

## 1. Introduction

*Lyssavirus rabies* (RABV) (*Mononegavirales: Rhabdoviridae: Alpharhabdovirinae*) is a negative-sense single-stranded RNA virus that causes rabies, a fatal encephalitis with 59,000 human deaths annually, mainly in the Asia and Africa [1,2,3]. The RABV genome comprises five genes encoding five proteins: nucleoprotein, phosphoprotein, matrix protein, glycoprotein and large protein [2]. The nucleoprotein is the most conserved of all lyssaviruses [4,5], while the glycoprotein is involved in the definition of the RABV phenotype regarding pathogenicity and neuroinvasiveness [6,7]. Therefore, nonsynonymous substitutions in these proteins can result in either catastrophic or advantageous consequences for the virus.

It is established that RNA viruses present high mutation rates due to low RNA polymerase replication fidelity [8]. Consequently, a variety of genomes can coexist, termed mutant spectra, forming a population structure known as a viral quasispecies [9], which is the core of RNA viruses’ adaptive strategy. In this context, under new microenvironmental conditions, the selection of the fittest populations, i.e., those with the highest replicative adaptability, might occur rapidly, conferring antiviral resistance, an affinity for a different receptor or even a greater virulence [10,11,12,13].

For instance, genetic changes that allow the adaptation of a given virus to a new host will result in an increase of fitness in this new environment [14]. The study at the molecular level can shed some light as to how this adaptation process occurs and can help to determine the molecular signatures related to the virulence.

Virulence markers or virulence gene signatures can be identified through the detection of nucleotide substitutions in specific portions of the genome, which can be related to phenotypic changes if they can affect the function and structure of viral proteins, and the substitutions that confer some degree of advantage to the virus can be fixed in a given population [15]. For instance, nucleotide substitutions could be involved in the affinity of viruses toward alternative attachment receptors in the host cell, such as in the case of RABV and heparan sulfate [16].

Without selective pressures, the mutant spectra manifest randomly. However, when a pressure is present, the evolution is constrained by this changing condition, as illustrated by the immune response, which can induce the selection of mutants able to overcome the immune effector mechanisms, and the greater the genetic diversity of a given viral population, the greater the selection potential of these mutants [17,18,19,20]. The selective pressure by neutralizing antibodies, for instance, is a well-established phenomenon that can affect viral evolution within a host and has been reported for a variety of viruses [21,22,23,24,25,26].

The investigation reported herein was designed to survey gene signatures on the nucleoprotein (N) and glycoprotein (G) genes that might be involved in virulence modulation during the in vitro evolution of the RABV lineages after serial passages in a neuronal cell system with or without the presence of neutralizing antibodies based on (a) replicative fitness, (b) in vivo neurotropism and (c) protein structure and dynamics.

## 2. Materials and Methods

### 2.1. Viruses and Cells

Five RABV isolates were selected after screening at the Laboratory of Diagnostics of Zoonosis and Vector-borne Diseases (LabZoo/DVZ/COVISA/SMS/PMSP) as a result of the Rabies Surveillance Program and represent the strains currently circulating in the State of São Paulo, Brazil (Table 1). A dog isolate (IP3629/11), kindly provided by the Pasteur Institute, Brazil, was also included in the analysis. All isolates were passed once through a mouse brain and then used for downstream assays.

Neuroblastoma cells (Neuro-2a—ATCC CCL-131) were cultivated in Eagle’s Minimum Essential Medium (EMEM) with Earle’s salts (Sigma-Aldrich, St. Louis, MO, USA) supplemented with 10% fetal bovine serum (FBS) (Bionutrientes, Taciba, SP, Brazil), 0.03% of nonessential amino acids (Sigma-Aldrich) and 1% penicillin–streptomycin (10,000 U/mL) (Sigma-Aldrich) at 37 °C in a 5% CO_2_ incubator.

### 2.2. Serial Passages in Neuro-2a Cell Line

For each isolate, a 20% viral suspension (*v*/*v*) in phosphate-buffered saline (PBS) supplemented with 1% penicillin–streptomycin (Sigma-Aldrich) and 2% FBS (Sigma-Aldrich) was prepared from a mouse brain after maceration The suspension was centrifuged at 3900× *g* for 10 min at 4 °C. Then, 0.5 mL of the supernatant was inoculated into 1 mL of Neuro-2a cells (8 × 10^5^ cells/mL) in 6-well cell culture plates (one for each isolate) and incubated for 1 h at 37 °C in a 5% CO_2_ incubator.

Next, 2 mL of fresh EMEM was added and the plates returned to incubation for another 72 h under the same conditions. The plates were then frozen at −70 °C for at least 24 h, thawed and the contents centrifuged at 3900× *g* for 10 min at 4 °C. One milliliter of the clarified lysate was used for the next passage, and an aliquot was used for RNA extraction, viral load quantification and Sanger sequencing of the N and G genes. A total of 10 passages were performed at the same conditions for each RABV isolate.

### 2.3. Serial Virus Neutralization (SVN) Assay in Neuro-2a Cell Line

The same steps described above for serial passages were performed for the SVN assay, with the exception that, prior to incubation with the cells, a preincubation of viral suspension with 0.05 UI/mL of horse anti-RABV F(ab’)2, kindly provided by the Ezequiel Dias Foundation was made for 1 h at 37 °C in a 5% CO_2_ incubator. Then, the protocol followed the steps used for the serial passage’s experiments (Section 2.2). Again, a total of 10 passages was performed for each RABV isolate at the same conditions, being the antibodies freshly added at each passage.

### 2.4. RNA Extraction and Reverse Transcription Reaction (RT)

RNA extraction and the RT reaction were performed as follows using 250 mL of the clarified lysates from the 10 passages for each isolate with or without antibodies, as well as supernatants from replication kinetics and competition assays.

RNA extraction was carried out using TRIzol (Thermo Fisher, Waltham, MA, USA) following the manufacturer’s instructions. The RT reaction was performed using the SuperScript VILO cDNA Synthesis Kit (Thermo Fisher) with random hexamers, also following the manufacturer’s instructions and stored at −20 °C until use. A volume of 2.5 µL of the cDNAs generated was used for PCR amplification of the N and G genes, as well as for viral load quantification by quantitative PCR (qPCR).

### 2.5. Primers for N and G genes Amplification

Primers for partial N gene amplification were designed using Primer-BLAST based on RABV sequences (GenBank Accession no. KM594038.1 and AB517659.1) and validated by the online tool PCR primers Stats (https://www.bioinformatics.org/sms2/pcr_primer_stats.html. Accessed on 16 December 2022) (Table 2). For complete G gene amplification, the primers used were described elsewhere [27].

### 2.6. Polymerase Chain Reaction (PCR) for N and G genes

All PCR reactions were performed using Taq Polymerase Platinum High Fidelity (Thermo Fisher) in a total volume of 25 µL, with High Fidelity Buffer 1X (600 mM Tris-SO_4_, pH 8.9 and 180 mM (NH_4_)2SO_4_); 1 U of Platinum Taq DNA Polymerase High Fidelity; 200 µM dNTP; 2 mM MgSO_4_; 0.5 µM of each primer and 2.5 µL of cDNA. DEPC-treated water and cDNA of the laboratory-adapted RABV isolate CVS (Challenge Virus Standard) were used as the negative and positive controls, respectively. Cycling parameters were 94 °C for 30 s, followed by 40 cycles of 94 °C for 15 s, 60 °C for 1 min and 68 °C for 2 min and a final stage of 68 °C for 5 min.

PCR products were confirmed by 1.5% agarose gel electrophoresis (Sigma-Aldrich) stained with SYBR safe (Thermo Fisher) and visualized using a UV transilluminator (Vilber Lourmat, Collégien, France).

### 2.7. Purification of PCR Products and Sanger Sequencing

The purification of PCR products was performed using ExoProStar 1-Step (GE Healthcare, Chicago, IL, USA), as instructed by the manufacturer. Bidirectional Sanger sequencing was accomplished using the BigDye Terminator v3.1 Cycle Sequencing Kit (Thermo Fisher) and the sequences generated in ABI-3500 (Applied Biosystems, Foster City, CA, USA). The sequences generated here were deposited in GenBank under accession numbers OP762196 to OP762335.

### 2.8. Sequence Editing and Analysis

Contig assembly for both the N and G genes was made using DNA Dragon—DNA Sequence Contig Assembler Software [28]. In order to identify nucleotide and amino acid polymorphisms, the contigs for each passage and for each isolate were aligned using the ClustalW multiple alignment tool in BioEdit Sequence Alignment Editor v. 7.0.3 [29].

### 2.9. Viral Titration

For each isolate, the clarified lysate corresponding to passages 1 and 10 were titrated using FAVN [30]. Briefly, Neuro-2a cells were seeded in 96-well culture plates (2x10^5^ cells/mL) and incubated at 37 °C and 5% CO_2_. Twenty-four hours later, the medium was aspirated, and each viral sample was diluted by a factor of 5 for a total of 10 dilutions, and 50 µL of each dilution was applied to the cell monolayer in triplicate. The plates were incubated at the same conditions for 1 h, and then, fresh medium was added, returning the plates for 72 h more of incubation. Next, the plates were fixed with 80% cold acetone and stained with anti-rabies conjugate (Merck KgaA, Darmstadt, HE, Germany). Every well with stained cells was considered positive, and the titer in TCID_50_/mL calculated by the Spearman-Karber method.

### 2.10. Replication Kinetics

The clarified lysate of passages 1 and 10 of each isolate were submitted to the replication kinetics assay at 0.01 MOI in Neuro-2a cells. The plates were incubated at 37 °C with 5% CO_2_, and aliquots of the supernatant were harvested at 16, 24, 48, 72 and 96 h post-infection and submitted for RNA extraction, RT reaction as previously described and viral load quantification by qPCR. Each sample was plated in quintuplicate, and the assay was performed three times, independently.

### 2.11. In Vivo Assay—Neurotropism Assessment and Ethics Statement

In order to assess how in vitro passages affect the RABV neurovirulence, the clarified lysate of passages 1 and 10 for each isolate were diluted to obtain 100 TCID_50_, adding up to 12 samples. For each sample, a group of five Swiss albino mice were anesthetized by halothane inhalation and submitted to intracerebral inoculation with 0.03 mL.

The mice were observed daily and euthanized as soon as nervous symptomatology was detected, such as agitation, spasms, aggressiveness, bristling hairs, abnormal gait, posterior limbs paralysis and kyphosis. The Central Nervous System (CNS) was extracted for diagnostic confirmation through direct immunofluorescence (DIF), as described elsewhere [31].

All in vivo testing was handled in accordance with good animal practice, as defined by the national animal welfare authorities, and was approved by the Ethic Committee on Animal Use of the School of Veterinary Medicine and Animal Science of the University of São Paulo (CEUA no. 5026010920).

### 2.12. Competition Assays

In order to assess the competition between the RABV populations at the first and the tenth passages in the cells as a proxy for the stability of the fitness, Neuro-2a cells were plated in 96-well plates at a concentration of 10^5^ cells/mL and incubated at 37 °C with 5% CO_2_ for 24 h. The cell monolayer was inoculated with a proportion of 1:1 using the clarified lysate of passages 1 and 10 at 0.1 MOI in triplicate. Each passage (1 and 10) was also tested separately for each isolate. The plates were incubated for 72 h in the same conditions above. Next, the supernatant of each well was submitted to 5 blind passages in Neuro-2a. All samples were submitted for RNA extraction, RT-PCR and Sanger sequencing for the G gene, as already described.

### 2.13. Quantification of Viral Load by qPCR

The quantification of the viral load in genome copies/mL was performed by qPCR through the absolute quantification method using a standard curve. qPCR was carried out using the primers JW-12 and N165-146 targeting the N gene [31] in a 20 µL reaction with 1 X Power SYBR^®^ Green (Thermo Fisher), 0.5 µM of each primer and 2.5 µL of cDNA. The StepOne^TM^ Real-time PCR System (Thermo Fisher) was used. The cycling parameters were 95 °C for 10 min followed by 40 cycles of 94 °C and 55 °C and 72 °C for 30 s. A melting curve was built after a stage of 95 °C for 15 s and a step of 60 °C for 1 min, with a gradual increase of the temperature of 0.3 °C/s up to 95 °C.

For normalization of the expression levels of the N gene, the β-actin gene was used as the endogenous control, as described elsewhere [32].

### 2.14. Three-Dimensional (3D) Modeling of G Protein, Optimization, Validation and Analysis

For homology modeling, the putative amino acid sequences of the G protein from passages 1 and 10 were submitted to SWISS-MODEL [33] using the templates for prefusion (PDB ID: 7U9G) and post-fusion (PDB ID: 6LGW) states. The obtained models were optimized by energy minimization with the ModRefiner algorithm [34]. For validation, the models were submitted to the online tool Structure Analysis and Verification server—SAVES v6.0; the Ramachandran plots generated for each model were assessed, those with more than 90% of the residues located in the most favorable regions being considered as a good quality model.

Model visualization and analyses were carried out using UCSF Chimera 1.15 [35] developed by the Resource for Biocomputing, Visualization, and Informatics at the University of California, San Francisco, with support from NIH P41-GM103311.

### 2.15. Molecular Dynamics Simulations

For performing molecular dynamics simulations, the GROMACS v. 2021.4-2 simulation package was used, with the OPLS-AA force field [36]. Protein structures were solvated in cubic boxes with 10 angstroms between the solute and box filled with SPC/E water molecules. The system was neutralized by adding ions Na^+^ or Cl^-^. For energy minimization, the steepest descent algorithm was used. System equilibration was performed with the NVT ensemble at 310 K, followed by the NPT ensemble for 100 ps. Berendsen thermostat coupling and Parrinello-Rahman pressure coupling were applied to keep the system at 310 K and 1 bar pressure, respectively. The particle mesh Ewald algorithm was applied for the electrostatic and Van der Waals interactions. The MD simulations were performed in 50 nanoseconds (ns), and the root mean square deviation (RMSD) and root mean square fluctuations (RMSF) were carried out also in GROMACS and analyzed in Xmgrace [37].

### 2.16. Molecular Docking

The 3D protein models were submitted to the ClusPro online server [38] for protein-protein docking, using the heparin module, as heparan sulfate is considered to be a secondary receptor for RABV [16]. The docking results were also visualized in UCSF Chimera. The 2D diagrams highlighting the ligand–receptor interactions were generated in Discovery Studio Visualizer, version 21.1.0.20298 (BIOVIA, San Diego, CA, USA).

### 2.17. Statistical Analysis

For the replication kinetics assay, the data were presented as the mean values ± SD of three independent experiments and were compared using two-way ANOVA with Bonferroni’s post-test. As for the in vivo assays, survival curves were estimated using the Kaplan–Meier method with Mantel–Cox test (log-rank test) for determining the statistical significance. GraphPad Prism 5.0 (GraphPad Software, San Diego, CA, USA) was used for the analysis.

## 3. Results

### 3.1. Amino Acid Substitutions during In Vitro Evolution of RABV Glycoprotein Gene Contribute to Increased Viral Loads in Neuronal Cell Line and Neurotropism In Vivo

To assess the emergence of RABV variants during 10 in vitro serial passages in Neuro-2a cells, the evolution of the N and G genes was evaluated. As expected, the N gene did not show polymorphisms, being conserved during the experiments. On the other hand, the G gene presented nonsynonymous nucleotide substitutions that led to putative amino acid substitutions in all the six field RABV isolates included in this study in the glycoprotein ectodomain and one of them specifically located in antigenic site III (Table 3).

The isolate from the Mexican free-tailed bat showed a substitution from serine to proline at amino acid site 165 at the 6th passage, which prevailed up to the 10th passage.

A mutation of glycine to glutamic acid was detected at the 3rd passage at amino acid site 186 for isolate 1069/18 (from a Great fruit-eating bat). The same amino acid site showed a glycine-to-arginine substitution at the 5th passage for three isolates: 1833/21 and 2193/21 (also from Great fruit-eating bats) and 2253/21 (from a white-eared opossum). In all cases, the substitution remained until the 10th passage.

Amino acid substitutions were also observed at site 188 from serine to phenylalanine at the 8th passage for isolate 1833/21 from a Great fruit-eating bat and from serine to proline at the 6th passage for isolates 2193/21 and 2253/21 from a Great fruit-eating bat and a white-eared opossum, respectively.

Lastly, an amino acid replacement at site 333 was detected for the dog isolate, from arginine to glutamine at the 6th passage, becoming dominant and lasting until the 10th passage.

To evaluate whether the substitutions observed in the putative sequences of the G protein affected the viral loads in vitro, a replication kinetics assay was performed comparing the 1st and 10th passages during 16, 24, 48, 72 and 96 h post-inoculation (p.i.) in Neuro-2a cells at 0.01 MOI. All variants from all isolates presented an increase in the viral load when compared to their wild types at all time points, reaching similar viral loads at 96 h p.i. (Figure 1).

A similar pattern was observed for the dog and the Mexican free-tailed bat isolates as the differences between the variant (10th cell passage) and wild type (1st cell passage), varied close to one order of magnitude at 24, 48, 72 and 96 h p.i. (Figure 1b,c, Great fruit-eating bat and white-eared opossum isolates (Figure 1a,d–f) presented a greater increase in virus loads between the variant and wild type passages, which reached four orders of magnitude for the isolate 1069/18 at 72 h p.i. (Figure 1a).

To assess the effect of these increased virus loads and amino acid substitutions on the neurovirulence, adult mice were intracranially inoculated with 100 TCDI_50_ of the wild type and variant of each isolate and observed for the manifestation of signs of rabies.

A decrease of 1, 5 and 6 days in the incubation period was observed between the wild type and the variant for the Mexican free-tailed bat, Great fruit-eating bat and white-eared opossum isolates, respectively (Figure 2a,c,f), all statistically significant. As for isolates 1833/21 and 2193/21, also derived from the Great fruit-eating bat (Figure 2d,e, respectively, the former showed a decrease in the incubation period between the wild type and variant, although not statistically significant, and the latter showed no difference in incubation time between the wild type and variant.

Conversely, the dog isolate showed a 5-day increase (Figure 2b) between the wild type and the variant, from 10 to 15 days, a statistically significant difference.

Taken together, the results suggest that amino acid substitutions in RABV glycoprotein after in vitro evolution of the field isolates can contribute to an increased fitness, as well as a higher neurotropism in vivo, when compared to the respective wild types.

### 3.2. G186E and G186R^+^S188P Glycoprotein Gene Variants Show Enhancement of Replicative Fitness In Vitro and Are Present in Natura

As one of the isolates from the Great fruit-eating bat (1069/18) and the isolate from the white-eared opossum (2253/21) presented a higher increase in the virus loads in vitro and decrease in the incubation period in mice, they were selected for the competition. The wild type and variant of both isolates were cultured at equal proportions (1:1) in Neuro-2a cell monolayers, with a total MOI of 0.1. Following the incubation period of 72 h, the supernatant of each well was used for five passages under the same conditions.

For the Great fruit-eating bat isolate, an adenine became dominant over a guanine at the second position in codon 186 already at the first passage (Figure 3a), which led to the substitution of codon GGG to GAG and the consequent amino acid substitution G186E. For the white-eared opossum isolate, a gradual guanine-to-adenine substitution at the first position of codon 186 (GGG to AGG) resulted in a G186R amino acid (Figure 3b). Thus, for both isolates, the variants became the dominant population regarding the G gene, outcompeting its wild type and suggesting that the G186E and G186R^+^S188P mutations contributed to the enhancement of in vitro fitness.

Eight sequences presenting the G186E substitution were found in GenBank, but three of them were related to serial passages of field isolates in mice (AB618032, AB618033 and AB618034). The other five belonged to field RABV isolates from Great fruit-eating bats (AB496693 and AB496685) and dogs (M81058, M81059 and M81060).

The G186R substitution was found in three sequences related to field isolates (KM594032, LT909527 and LT909549) in the laboratory-adapted strain CVS-11 (FJ979833) and in the EPHVAC vaccine strain (JQ944709). In addition to RABV, an arginine at position 186 was also identified in two other lyssaviruses, *Lyssavirus shimoni* (NC_025365 and GU170201) and *Lyssavirus caucasicus* (NC_025377, EF614258 and MZ501949), after a search of complete genome sequences of other lyssaviruses in GenBank.

Substitutions in position 188 were found in three sequences (JQ685963, GU937034 and KM594033) also corresponding to field isolates, but the substitutions were to phenylalanine or tyrosine instead of proline.

These data suggest that G protein variants G186E and G186R^+^S188P lead to an increased fitness during competition and a possible role of amino acid residues 186 and 188 in RABV virulence modulation.

### 3.3. Amino Acid Substitutions G186E and G186R^+^S188P in the G gene Affects Glycoprotein Structure and Dynamics

Three-dimensional (3D) homology models were built using the putative glycoprotein sequences of the wild types and variants of one of the Great fruit-eating bat isolates (1069/18) and a white-eared opossum isolate (2253/21), and the sites with mutations were depicted. Positions 186 and 188 are in the glycoprotein ectodomain—more precisely, in a domain linker between the Pleckstrin Homology Domain (PHD) and Fusion Domain (FD), the latter comprising the two fusion loops (Figure 4a,b).

In order to search for possible differences in protein behaviors between wild types G186 and G186^+^S188 and their respective variants G186E and G186R^+^S188P, molecular dynamics simulations were performed. The root mean square deviation (RMSD) for the trajectories of Cα backbone were calculated for the wild type and variants, high values being indicative of higher deviations and, therefore, a more unstable protein structure.

Deviations in the wild type G186 protein structure were observed between 10 and 20 ns of the simulation and again between 25 and 35 ns. After 35 ns, the RMSD trajectory values remained constant until the end of the simulation for both protein structures. The mutant G186E showed a protein structure more stable when compared to its wild type G186 with an average RMSD value of 0.52 nm against 0.66 nm (Figure 5a).

Regarding mutant G186R^+^S188P and its wild type G186^+^S188, the RMSD trajectory showed a higher deviation for the wild type between 20 and 35 ns of the simulation. After that, both structures remained similarly stable. Comparing RMSD values, wild type G186^+^S188 was slightly less stable than mutant G186R^+^S188P, with averages of 0.64 nm and 0.61 nm, respectively (Figure 5b).

Another measure applied to the data was the root mean square fluctuations (RMSF), which allow the assessment of protein flexibility during the simulation time.

The RMSF plot for mutant G186E and its wild type showed a higher flexibility for the mutant protein structure in fusion loop 1 (amino acids 71 to 77), residue 74 being the most flexible, with a value of 0.88 nm compared to 0.65 nm for the wild type (Figure 5c).

A similar result was verified for the mutant G186R^+^S188P and its wild type, with an increased flexibility of fusion loop 1 for the mutant protein structure. Again, residue 74 was more flexible for the mutant, with a RMSF value of 0.94 nm compared to 0.82 for the wild type. However, fusion loop 2 (amino acids 119 to 124) presented an even higher fluctuation for mutant G186R^+^S188P, with residue 122 being the most flexible, with a RMSF value of 1.02 for the mutant compared to 0.51 for the wild type (Figure 5d).

Taken together, these data indicate more stable mutant protein structures with increased flexibility in the fusion peptides, possibly affecting the endosome fusion of the virion and host cell membrane.

### 3.4. Mutant G186R Shows Interaction with Heparin In Silico

It was pointed out in the literature that an arginine at position 186 of the glycoprotein could be involved in the interaction with heparin in a laboratory-adapted RABV strain [16]. As three isolates of the present study showed a substitution to arginine in this position after cell passages, we decided to investigate whether this was true for the variants generated here. Therefore, the 3D model obtained from the putative glycoprotein sequence of the variant (10th passage in Neuro-2a cells) of isolate 2253/21 was submitted to molecular docking with heparin. The mutation G186R contributed to the formation of a positively charged pocket of amino acids that allowed the interaction with heparin, which is negatively charged (Figure 6a,b), a further indication that the 186 of RABV glycoprotein could act as a ligand for heparin. The 2D diagram showed a conventional hydrogen bond formed between heparin and R186 (Figure 2c).

### 3.5. Mutation in G Gene Can Promote Escape from Neutralizing Antibodies in a Field RABV Isolate

The serial virus neutralization (SVN) experiments showed that all RABV isolates except the dog one were completely neutralized by horse anti-RABV F(ab’)2 after ten neutralization rounds in Neuro-2a cells (Figure 7).

As shown in the previous sections, an R333Q substitution was detected in the G protein of the dog isolate, which remained until the 10th passage, suggesting that this substitution could be involved in escaping from neutralizing antibodies.

## 4. Discussion

In this study, six field RABV isolates were submitted to 10 serial passages in a neuronal cell line and mouse inoculation to test the hypotheses that (a) RABV evolution in cell cultures will lead to virulence modulation, allowing for the identification of virulence gene signatures, and (b) the introduction of neutralizing antibodies as a selective pressure could lead to the emergence of escape mutants.

As expected, all isolates presented an improved adaptation to the Neuro-2a cell line during in vitro evolution, as well as increased neurotropism in vivo, since all the experiments were carried out in the same host and in a controlled microenvironment, thus allowing the achievement of an optimal fitness [39].

At the molecular level, no substitutions were detected in the N gene for any of the isolates during in vitro evolution. For the G protein, the substitutions S165P, G186E, G186R, S188F, S188P and R333Q were identified, all of them in the ectodomain, R333Q being located in antigenic site III of the glycoprotein. These findings were in agreement with a previous study that reported great stability of the N gene and nucleotide substitutions in the glycoprotein ectodomain during the in vitro evolution of RABV [40].

These G protein substitutions became dominant along passages in the Neuro-2 cells, altering the consensus sequence for the G gene not only from sub-consensual variants already present in the initial viral population but also from the mutations that emerged de novo during the serial passages. Microenvironmental changes after passages in different hosts trigger a quick response from the G gene, which suggests that RABV glycoprotein is under a weak positive selective pressure, at least when compared to the nucleoprotein [41,42].

It is of note that the isolate derived from the white-eared opossum is phylogenetically close to the Great fruit-eating bats’ RABV lineage, to which isolates 1069/18 belong. It is known that the N and G gene signatures are found for a variety of RABV lineages from different hosts [43,44,45,46,47]. The G186E and G186R^+^S188P detected in the glycoprotein of these two isolates after 10 cell passages that resulted in increased neurovirulence in mice are in a region of the glycoprotein ectodomain related to RABV pathogenicity between amino acids 164 to 303, while amino acids 113, 164 and 254 are related to low neutralization by antibodies [48,49,50].

Amino acid residues at positions 186 and 188 also map to a domain linker between PHD and FD. Domain linkers are important, as they act in the maintenance of cooperative interactions between domains and favor the folding of proteins with multiple domains [51].

The molecular dynamics simulations of mutants G186E and G186R^+^S188P resulted in more fluctuations in the fusion loops, which comprehends amino acids 71 to 77 and 119 to 124 [52]. Conformational flexibility in the proteins allows a better adaptability to microenvironmental changes [53]. In the case of this study, the mutations mentioned above could have led to changes in the conformational movements of the domain linker that unites the PHD and the FD.

The RABV glycoprotein at pHs 8 and 6.5 shows significant differences in folding: the acidification of endosomal pH triggers conformational changes in the protein, which assumes its active state, initiating membrane fusion [54,55,56,57]. These changes cause a 180° rotation of the glycoprotein and circa a 59° rotation in the FD. This last rotation occurs due to conformational changes in the domain linker in which residues 186 and 188 are located. The post-fusion state of the glycoprotein corresponds to the stretching of the protein, exposing the fusion loops in the direction of the cell host membrane [54]. The more open conformation of the mutant glycoproteins could implicate in a more effective exposure of the fusion loops, that presented a higher fluctuation during molecular dynamics simulations, which would result in a more efficient fusion with endosomal membrane.

The G186R substitution also found in wild RABV as depicted from GenBank sequences, besides resulting in higher fitness in vitro and higher neurotropism in vivo, also facilitated the interaction with heparin in silico. Laboratory-adapted RABV strains can develop the ability to interact with other host cell receptors, including those not found in neuronal cells [58].

Heparin, and also heparan sulfate, are glycosaminoglycans or GAGs, have been reported to interact with a variety of viruses such as cytomegalovirus, dengue virus, sindbis, vaccinia, respiratory syncytial virus, maedi-visna virus, adenovirus and rhabdoviruses in general [59,60,61,62,63,64,65,66]. For flavivirus, the interaction with heparan sulfate is of low affinity, and the consequence would be the enhancement of viral particles in the surface of the host cell, improving attachment to other receptors with higher affinity [67].

In RABV, heparan sulfate could be an attachment factor rather than a cell receptor, acting as a secondary receptor, as shown for the herpesvirus, pseudorabies virus [16,68]. Thus, mutants in the quasispecies could be selected for their ability for attachment to alternative host cell receptors [11].

Regarding the virus neutralization assays, no substitutions were detected in both N and G genes for all the isolates, except the dog isolate, which showed the R333Q substitution fixed up to the tenth passage. Analysis of viral load of this isolate revealed a great stability during the ten passages, even after the predominance of R333Q. A possible explanation is that this mutation, which took place in an antigenic site, was fixed precisely because of the selective pressure exerted by the antibodies.

Amino acid substitutions at position 333 from arginine or lysine to glutamine, isoleucine, glycine, or methionine in mutants derived from RABV adapted isolates capable of escaping from neutralizing monoclonal antibodies would result in loss of virulence in adult mice after intracranial inoculation [69,70,71,72]. The cultivation of the adapted isolate SAD Bern (derived from the ERA isolate) in the presence of monoclonal antibodies, which are capable of specific attachment to glycoprotein antigenic site III, led to the selection of mutants capable of escaping neutralization in cell culture. These mutants presented a substitution in position 333, as well as the SAG2 isolate, derived from SAD Bern [73,74].

Biophysical changes after the R333Q substitution in the glycoprotein, from a positively charged to a neutral amino acid, could affect the interaction with antibodies. Antibodies capable of interacting with antigenic site III of RABV glycoprotein would act blocking the antigenic site, which in turn would not be able to attach to p75NTR, negatively charged receptor for RABV [54,69,75], leading to the lower in vivo virulence of the dog isolate after 10 cell passages.

It is noteworthy that a sub-dose of antibodies did not extinguish RABV population after the first passage, which would serve as reinforcement of the need for maintaining post-exposure prophylaxis protocols with more than one dose.

This study had some limitations that must be highlighted for a better understanding of their implications. In the case of the in vivo assays, a possible reversion to the wild type in mice was not evaluated, which could be addressed by sequencing the G gene after mice inoculation. Another limitation is that intracranial inoculation does not allow the differentiation between neurotropism, i.e., ability to infect neural cells, from neuroinvasiveness, i.e., ability to migrate from non-neural to neural cells, which could be investigated by using a peripheral route of inoculation, such as the intramuscular one.

Another issue is that Sanger sequencing directly from PCR products does not enable a full observation of the mutant spectra and, therefore, of the evolution from the perspective of quasispecies, which could be resolved by deep sequencing. Finally, the understanding of the individual role of each mutation needs to be investigated, which could be accomplished through reverse genetic experiments.

In conclusion, amino acids at positions 186 and 188 of the glycoprotein are virulence factors of *Lyssavirus rabies* and site 186 specifically might allow the attachment to heparan as a secondary cell receptor, while polymorphism at position 333 might allow the selection of escape mutants under suboptimal neutralizing antibodies titers.

## Figures and Tables

**Figure 1 pathogens-11-01556-f001:**
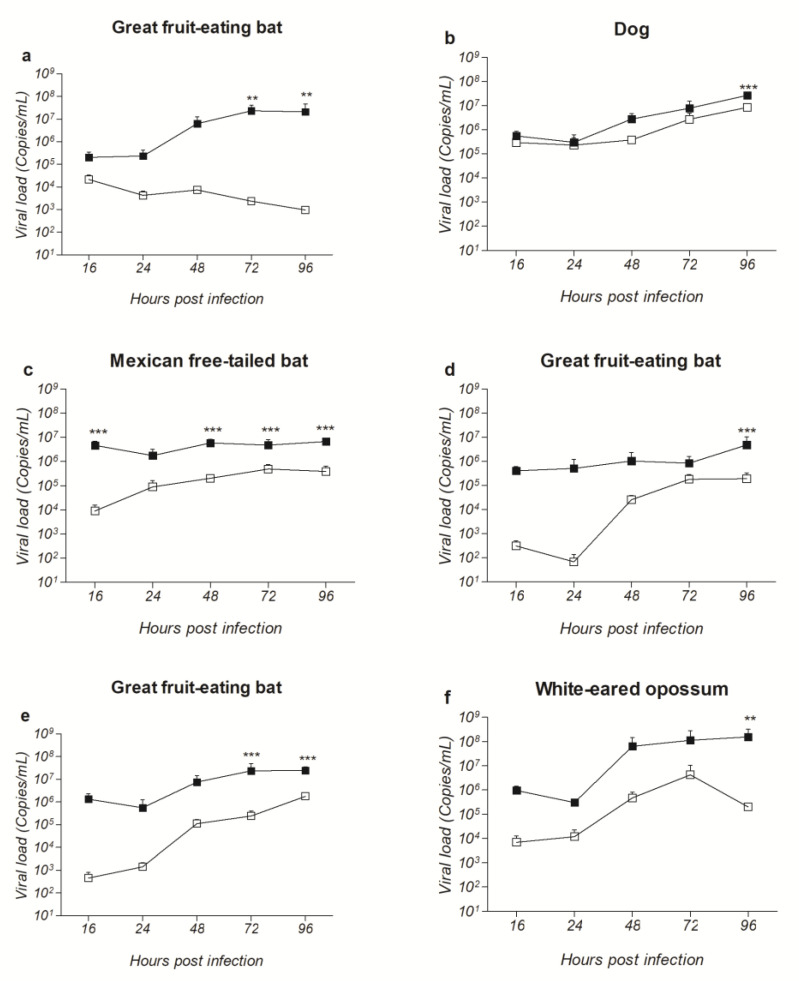
Replication kinetics of the wild type (1st cell passage, white squares) and variants (10th cell passage, black squares) of field RABV isolates after 16, 24, 48, 72 and 96 h post-infection in a neuronal cell line. Viral loads obtained by qPCR from culture supernatants, shown as log10 genome copies/mL. (**a**) Great fruit-eating bat (Isolate 1069/18). (**b**) Dog (Isolate IP3629/11). (**c**) Mexican free-tailed bat (Isolate 203/21). (**d**) Great fruit-eating bat (Isolate 1833/21). (**e**) Great fruit-eating bat (Isolate 2193/21). (**f**) White-eared opossum (Isolate 2253/21). Data represent the mean ± SD of three independent experiments. ** (*p* < 0.01) and *** (*p* < 0.001) comparing the wild type and the respective variant.

**Figure 2 pathogens-11-01556-f002:**
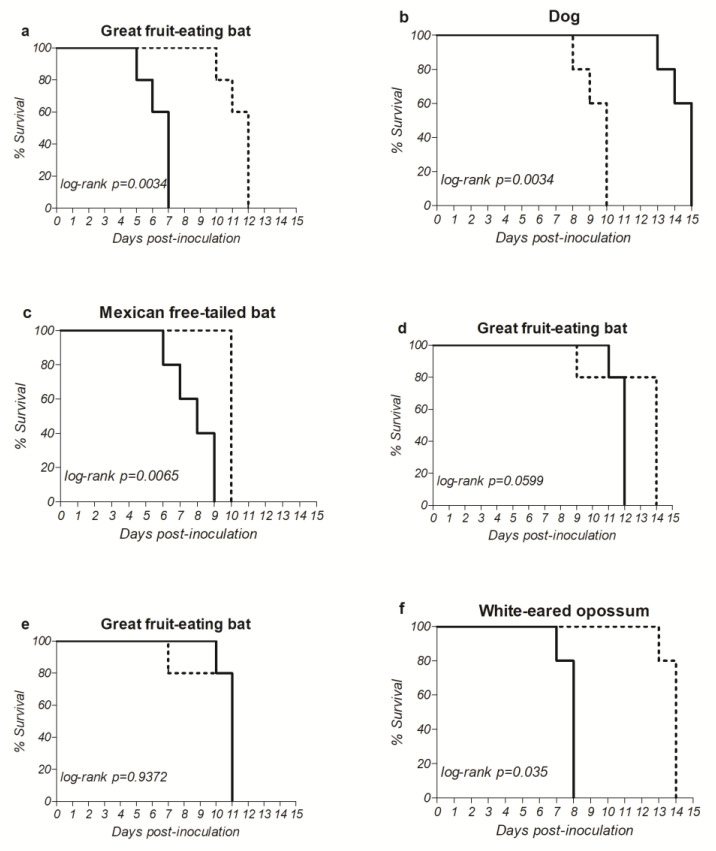
Kaplan–Meier curves. The results represent the survival after the intracranial inoculation of adult mice (five mice per sample) with the wild type (dotted lines) and variant (black lines) of field RABV isolates. (**a**) Great fruit-eating bat (Isolate 1069/18). (**b**) Dog (Isolate IP3629/11). (**c**) Mexican free-tailed bat (Isolate 203/21). (**d**) Great fruit-eating bat (Isolate 1833/21). (**e**) Great fruit-eating bat (Isolate 2193/21). (**f**) White-eared opossum (Isolate 2253/21). *p* < 0.05 comparing the incubation period of the wild type and the respective variant.

**Figure 3 pathogens-11-01556-f003:**
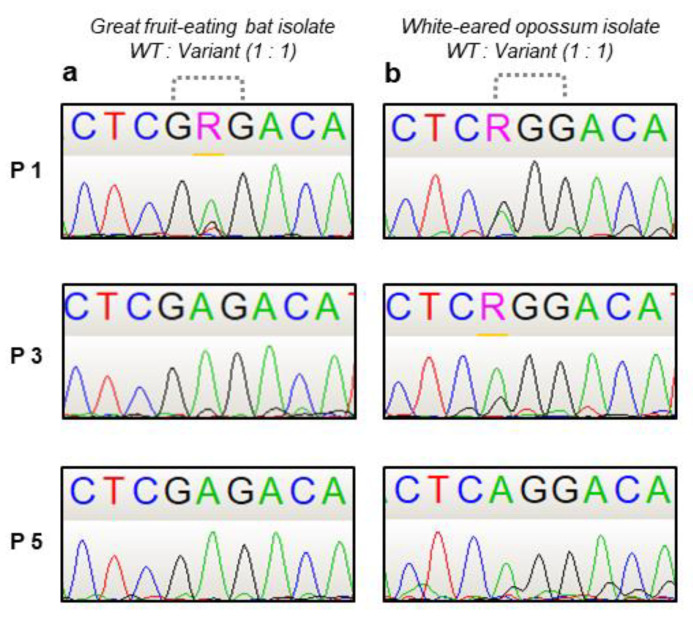
Sanger sequencing chromatograms of a fragment of the G gene from the wild type and variants of RABV isolates during the competition assay. Dotted dashes highlight the mutated codon. Figure depicts passages 1, 3 and 5. (**a**) Great fruit-eating bat isolate (1069/18). (**b**) White-eared opossum isolate (2253/21). R represents a purine (adenine or guanine).

**Figure 4 pathogens-11-01556-f004:**
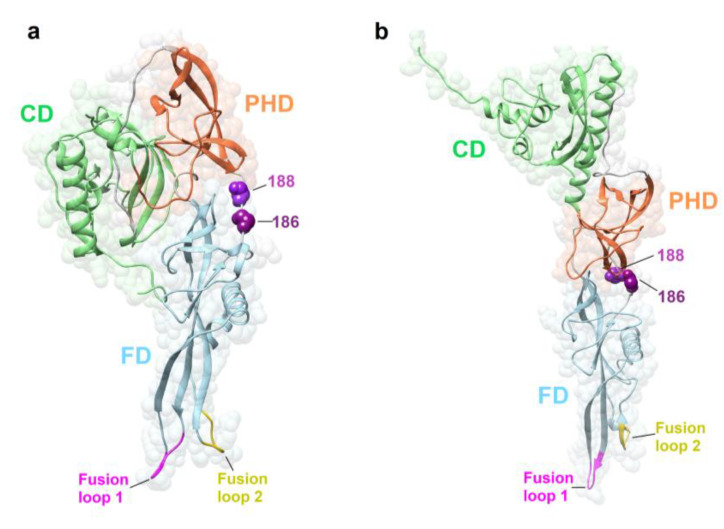
Three-dimensional model of the RABV glycoprotein. (**a**) Prefusion glycoprotein state. (**b**) Post-fusion glycoprotein state. Amino acid sites 186 and 188 located at the domain linker and the two fusion loops are highlighted. CD, Central domain; PHD, Pleckstrin Homology domain; FD, Fusion domain.

**Figure 5 pathogens-11-01556-f005:**
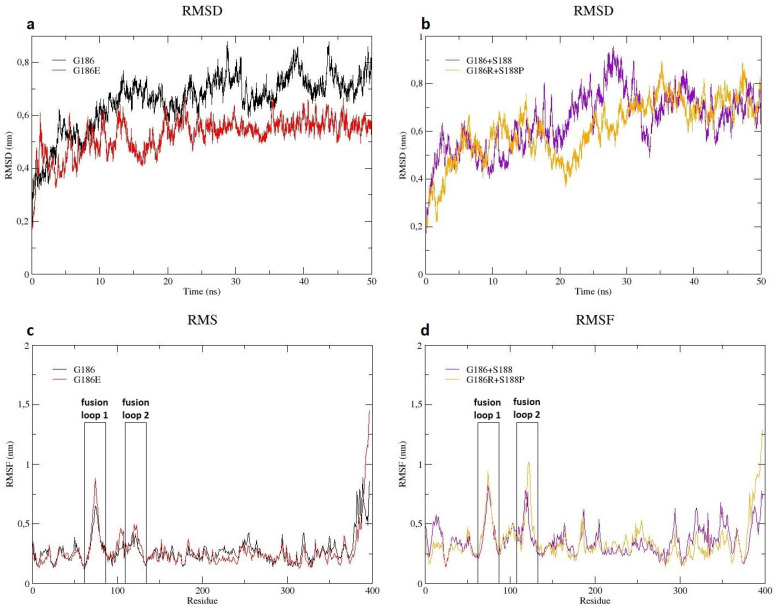
Analysis of the MD simulations. (**a**) Backbone RMSD analysis of wild type G186 and the respective variant G186E during 50 ns of simulation. (**b**) Backbone RMSD analysis of wild type G186^+^S188 and the respective variant G186R^+^S188P during 50 ns of simulation. (**c**) RMSF profile for wild type G186 and the respective variant G186E. (**d**) RMSF profile for wild type G186^+^S188 and the respective variant G186R^+^S188P.

**Figure 6 pathogens-11-01556-f006:**
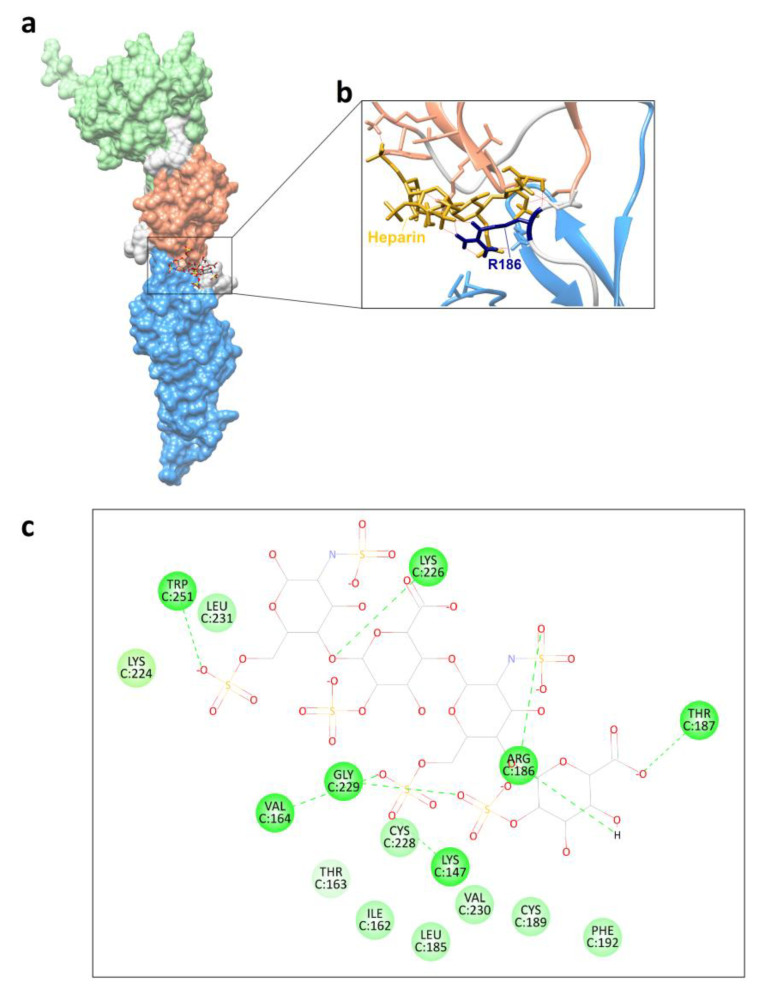
Molecular docking results with heparin. (**a**) Three-dimensional model of mutant glycoprotein G186R^+^S188P with heparin. (**b**) Emphasis on the interaction of heparin (golden) and R186 (navy blue). (**c**) Two-dimensional diagram showing the glycoprotein amino acids involved in the hydrogen bond (darker green and dotted lines) and van der Waals interactions (lighter green) with heparin.

**Figure 7 pathogens-11-01556-f007:**
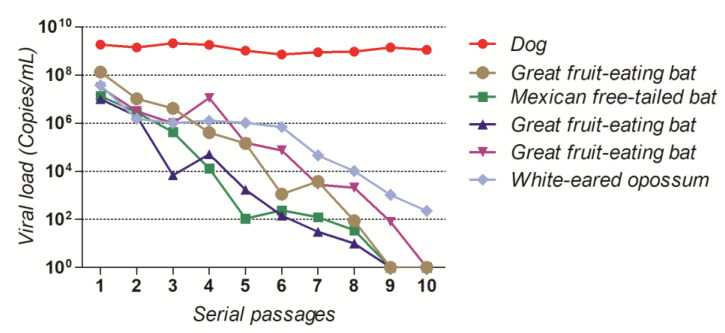
Viral load in clarified lysate after 10 rounds of virus neutralization assay in Neuro-2a cells with 0.05 UI/mL of horse anti-RABV F(ab’)2 for six of the field RABV isolates included in this study.

**Table 1 pathogens-11-01556-t001:** Field RABV isolates selected for this study.

Identification	Host Species	Common Name	Variant
1069/18	*Artibeus lituratus*	Great fruit-eating bat	Bat
IP3629/11	*Canis lupus familiaris*	Dog	Cosmopolitan dog
203/21	*Tadarida brasiliensis*	Mexican free-tailed bat	Bat
1833/21	*Artibeus lituratus*	Great fruit-eating bat	Bat
2193/21	*Artibeus lituratus*	Great fruit-eating bat	Bat
2253/21	*Didelphis albiventris*	White-eared opossum	Bat

**Table 2 pathogens-11-01556-t002:** Primers designed for N gene amplification.

Primer	Sense	Sequence	Size	Position *
N55	F	5′ ATGTAACACCTCTACAATGG 3′	842 bp	55–74
N879 **	R	5′ CAGGCTCGAACATTCTTC 3′	842 bp	879–896
N878 ***	R	5′ CCCGGGCTCGAACATTCTTCT 3′	844 bp	878–898
N750	F	5′ GCACAGTTGTCACTGCTT 3′	793 bp	750–767
N1525B **	R	5′ GCACTTGGGCTGACAAAA 3′	793 bp	1525–1542
N1525D ***	R	5′ TGCACTAGGATTGACAAAG 3′	793 bp	1525–1542

* Position numbered according to RABV isolate Pasteur Virus (PV) sequence (GenBank Accession no. M13215). ** For amplification of bat-derived samples. *** For amplification of dog-derived samples. F—forward primer; R—reverse primer.

**Table 3 pathogens-11-01556-t003:** Nonsynonymous nucleotide substitutions in the glycoprotein gene and putative amino acid substitutions during ten serial passages of field RABV isolates in the neuronal cell line.

Isolate		Substitution				
	Host	nt	Position *	aa	Position *	Passage no.
1069/18	Great fruit-eating bat	GGG–GAG	613	G–E	186	3
IP3629/11	Dog	CGG–CAG	1055	R–Q	333	6
203/21	Mexican free-tailed bat	TCC–CCC	550	S–P	165	6
1833/21	Great fruit-eating bat	GGG–AGG	613	G–R	186	5
		TCT–TTT	620	S–F	188	8
2193/21	Great fruit-eating bat	GGG–AGG	613	G–R	186	5
		TCT–CCT	619	S–P	188	6
2253/21	White-eared opossum	GGG–AGG	613	G–R	186	5
		TCT–CCT	619	S–P	188	6

* Position numbered according to RABV isolate Pasteur Virus (PV) sequence (GenBank Accession no. M13215); nt—nucleotide; aa—amino acid.

## Data Availability

The sequences generated in the present study were submitted to GenBank with accession numbers OP762196–OP762335.

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
