# Peer review of "Evolution of Rabies Virus Isolates: Virulence Signatures and Effects of Modulation by Neutralizing Antibodies"

_pathogens, 2022, doi:10.3390/pathogens11121556_

Round 1
Reviewer 1 Report
In this manuscript by Conselheiro et al., the authors investigated gene signatures that could be involved in virulence modulation. Overall, the manuscript is well written, methodology is clearly described with a comprehensive discussion and was very interesting to read.
My only comment is that the authors are confusing virus and species names throughout the manuscript and this should please be corrected as per the International Committee on Taxonomy of Viruses rules
Reviewer 2 Report
The manuscript Evolution of rabies virus isolates: virulence signatures and effects of modulation by neutralizing antibodies by J. A. Conselheiro and colleagues has interesting hypothesis and results and plenty of methods.
However, some corrections are necessary.
Introduction
Line 27: Rabies lyssavirus instead Lyssavirus rabies. Put RABV in brackets.
Material and Methods
Lines 66-70: if I understood correctly, the authors selected four (or five (if opossum is considered) but not six) RABV isolates from the collection of the Laboratory of Diagnostics of Zoonosis and Vector-borne Diseases. The dog isolate (IP3229/11) was kindly provided by the Pasteur Institute.
I do not understand the origin of the opossum isolate (2253/21). Please explain in more detail.
Do all RABV isolates from the collection of the Laboratory of Diagnostics of Zoonosis and Vector-borne Diseases belong to bats (and/or opossum)? Please emphasize this for easier understanding.
Table 1. I suggest introducing column with common species name (great fruit eating bat, dog…).
Lines 80-82: the authors should describe preparation of viral suspension from mouse brain in more detail. Why did the authors decide to pass the isolates once in the mouse brain instead of entirely on the cell line?
Line 88: the authors should specify amount used for RNA extraction, viral load quantification and Sanger sequencing of N and G genes.
Line 91: I am confused why the authors used the term Serum Virus Neutralization assay when they did not use serum.
Table 2. is a bit confusing. It should be better organized. Do bats and dog have the same F primer but different R primers? Column Position need some editing.
Line 137: put space
Lines 162-165: the authors stated that they included six isolates (see earlier comment Lines 66-70). Here the authors stated, ‘the clarified lysate of passages 1 and 10 for each isolate were diluted to obtain 100 TCID50', meaning 12 isolates? However, they inoculated only five mice. Please explain discrepancy, which isolates and passages you used for inoculation.
Results
Line 262: post-inoculation (p.i.)
Lines 279-281: please check earlier comment (Lines 162-165).
Line 282: Please check ‘A decrease of 2...'. From Figure 2c it is possible to observe decrease of 1 day.
Lines 284-285: '...2193/21 (Figure 2e) also showed a decrease in the incubation period between wild type and variant...'. I do not see decrease on the Figure 2e, there is no difference between wild type and variant. Please check.
Lines 287-288: there is a mismatch between the text (…from 9 to 14 days...) and the Figure 2b (from 10 to 15 days), please check.
Figure 1. I think there is an error in the number of **/*** in the legend. Please check.
Figure 2. ‘…five mice per group...' I do not understand what the authors considered as a group.
Figure 3. What is the meaning of R on the Figure 3? Please put explanation in the legend.
Lines 402-404: earlier (Lines 257-259) authors stated that an amino acid replacement at site 333 was detected for the isolate IP3629/11, derived from a dog, from arginine to glutamine at the 6th passage. Here the authors state that this happend in the 7th passage.
Discussion
Line 424: no need for a Latin name. How did the authors conclude about phylogenetic relationship between opossum and bat RABV lineage?
Lines 456-461: I suggest deleting this paragraph
Line 466: serum neutralization assays – see earlier comment
Reviewer 3 Report
The manuscript by Conselheiro describing the virulence of different rabies viruses is an interesting study that evaluates several different aspects of rabies virus and how each contribute to its pathogenesis. Although this would be a good addition to the field of virology, it misses some important points. For instance, they include an opossum isolate, which is interesting in itself as opossums are one of the less susceptible species to rabies virus infection, which was not discussed or highlighted, but in this reviewers opinion, it deserves much more attention. Additionally, the study is evaluating the virus after growing in mouse cells so increased adaptation to mice vs the wt is not surprising, yet not reviewed in the discussion.
It would be valuable to replace the isolate ID with the common animal name throughout the paper and tables/figures. It was difficult to compare the different variant characteristics in the tables and text when it requires flipping back and forth to table 1 to determine if the bat variants are growing better or the dog …….
Figures- please look at all figures in black and white and determine if the colors you are using can be distinguished in black and white print.
Specific comments
Table 1. Please include species/common name/variant aka Canis lupus familiaris /dog/ cosmopolitan dog variant. Are the other 4 bat variants? Would suggest including a phylogenic tree to illustrate relatedness of the variants.
2.2. Was the viral suspension spun down and the clarified portion used for the inoculate? Please expand on the methods
Line 246. Would suggest adding a space or two to separate this paragraph from the footnotes
Line 258. Dog is included here which is helpful, but this should be done throughout the text.
Line 324. Two other lyssas are mentioned Shimoni and WCBV which have very few isolates. How many of these isolates were you about to use for comparison to your isolates? Is it statistically significant?
Line 222 and 379-380. Would suggest adding a line or two to the introduction explaining why you are including heparin in this experiment.
Line 424. It is stated that the variant derived from the opossum is close to the bat RABV. Does this mean it is a spill over case? Or is this variant adapted for opossums? Is this a circulating opossum variant?
Paragraph beginning with 456. Rabies and its relationship with heparin is mentioned but what about other rhabdoviruses? There are several other rhabdoviruses that infect animals (VSV) fish and plants.
Line 48. P75NTR is mentioned but what about all the other rabies virus receptors?
Line 486. What is considered a sub dose? Please provide a reference for this information/paragraph
Paragraph staring with 496. Why wasn’t WGS part of this project?
